# Isoniazid Concentration and *NAT2* Genotype Predict Risk of Systemic Drug Reactions during 3HP for LTBI

**DOI:** 10.3390/jcm8060812

**Published:** 2019-06-06

**Authors:** Meng-Rui Lee, Hung-Ling Huang, Shu-Wen Lin, Meng-Hsuan Cheng, Ya-Ting Lin, So-Yi Chang, Bo-Shiun Yan, Ching-Hua Kuo, Po-Liang Lu, Jann-Yuan Wang, Inn-Wen Chong

**Affiliations:** 1Department of Internal Medicine, National Taiwan University Hospital, Hsinchu Branch, Hsinchu 30059, Taiwan; sheepman1024@gmail.com; 2Division of Pulmonary and Critical Care Medicine, Department of Internal Medicine, Kaohsiung Medical University Hospital, Kaohsiung 80708, Taiwan; 990325kmuh@gmail.com (H.-L.H.); cmhkmu@gmail.com (M.-H.C.); chong@kmu.edu.tw (I.-W.C.); 3Graduate Institute of Medicine, College of Medicine, Kaohsiung Medical University, Kaohsiung 80708, Taiwan; d830166@gmail.com; 4Kaohsiung Municipal Ta-Tung Hospital, Kaohsiung 80145, Taiwan; 5School of Pharmacy, College of Medicine, National Taiwan University, Taipei 10050, Taiwan; shuwenlin@ntu.edu.tw (S.-W.L.); ytljolly@gmail.com (Y.-T.L.); kuoch@ntu.edu.tw (C.-H.K.); 6School of Medicine, College of Medicine, Kaohsiung Medical University, Kaohsiung 80708, Taiwan; 7Departments of Respiratory Therapy, Kaohsiung Medical University, Kaohsiung 80708, Taiwan; 8Institute of Biochemistry and Molecular Biology, National Taiwan University Medical College, Taipei 10051, Taiwan; vagrantlin@hotmail.com (S.-Y.C.); bsyan@ntu.edu.tw (B.-S.Y.); 9Department of Laboratory Medicine, Kaohsiung Medical University Hospital, Kaohsiung 80756, Taiwan; 10Department of Internal Medicine, National Taiwan University Hospital, Taipei 10002, Taiwan

**Keywords:** isoniazid, *N-acetyltransferase 2*, rifapentine, latent tuberculosis infection, systemic drug reaction

## Abstract

Weekly rifapentine and isoniazid therapy (known as 3HP) for latent tuberculosis infection (LTBI) is increasingly used, but systemic drug reactions (SDR) remain a major concern. Methods: We prospectively recruited two LTBI cohorts who received the 3HP regimen. In the single-nucleotide polymorphism (SNP) cohort, we collected clinical information of SDRs and examined the *NAT2*, *CYP2E1*, and *AADAC* SNPs. In the pharmacokinetic (PK) cohort, we measured plasma drug and metabolite levels at 6 and 24 h after 3HP administration. The generalised estimating equation model was used to identify the factors associated with SDRs. Candidate SNPs predicting SDRs were validated in the PK cohort. A total of 177 participants were recruited into the SNP cohort and 129 into the PK cohort, with 14 (8%) and 13 (10%) in these two cohorts developing SDRs, respectively. In the SNP cohort, *NAT2* rs1041983 (TT vs. CC+CT, odds ratio [OR] [95% CI]: 7.00 [2.03–24.1]) and *CYP2E1* rs2070673 (AA vs. TT+TA, OR [95% CI]: 3.50 [1.02–12.0]) were associated with SDR development. In the PK cohort, isoniazid level 24 h after 3HP administration (OR [95% CI]: 1.61 [1.15–2.25]) was associated with SDRs. Additionally, the association between the *NAT2* SNP and SDRs was validated in the PK cohort (rs1041983 TT vs. CC+CT, OR [95% CI]: 4.43 [1.30–15.1]). Conclusions: Isoniazid played a role in the development of 3HP-related SDRs. This could provide insight for further design of a more optimal regimen for latent TB infection.

## 1. Introduction

Tuberculosis (TB) remains one of the deadliest infectious diseases, with an estimated 10.0 million new cases and 1.6 million deaths in 2017 [1]. The World Health Organization (WHO) has set the goal of eliminating TB as a public health problem, aiming to achieve 90% and 95% reductions in the TB incidence and number of TB deaths by 2035 [2]. Latent tuberculosis infection (LTBI), a status of persistent immune response to stimulation by *Mycobacterium tuberculosis* antigens without clinically manifest active TB [3], has a 10% risk of progressing to active TB [4] and has thus emerged as a critical target for improving TB control and elimination [5]. The importance of targeting LTBI toward TB control has extended from countries with low TB prevalence to TB-endemic areas [6,7]. LTBI treatment, therefore, is being advocated as a universal policy in TB control [6,7].

The effectiveness of LTBI programmes has long been limited. Real-world data published in 1999 and obtained from an inner-city population in Atlanta, Georgia in the United States revealed that only 27% of subjects who received isoniazid (INH) preventive therapy completed their treatment [8]. A 2016 meta-analysis including 748,572 subjects in 58 studies also found poor completion of LTBI programmes, with a 60% treatment completion rate [9]. With the introduction of rifapentine (RPT), a rifamycin with much longer half-life than rifampin, the duration of a modern preventive regimen termed 3HP comprising RPT and INH could be shortened to 12 doses administered weekly, with the completion rate approaching 90% [10,11,12]. Also, for the four LTBI regimens currently suggested by WHO, 3HP remained the one with the fewest total doses required [13]. Though 3HP is less hepatotoxic, 3.8% of those receiving 3HP experience systemic drug reactions (SDRs) [14], which usually, if not always, requires treatment interruption or termination. The risk of a severe adverse event (AE) and SDR are even higher among subjects > 35 years old [12,14]. 

To date, little has been discovered regarding the risk factors or predictors of SDRs due to 3HP therapy. In a pharmacokinetics study of RPT treatment in 35 TB patients during once-weekly continuation phase therapy, serious AEs were not linked with a higher area under the plasma concentration–time curve (AUC_0~∞_) of RPT [15]. Furthermore, no studies have reported on plasma INH levels after once-weekly INH treatment or the association between plasma INH level and AE development. Therefore, we conducted this study to determine the predictors of SDRs during 3HP therapy by measuring the plasma levels of drugs and their major metabolites and by genotyping the three key drug-metabolising enzymes.

## 2. Methods

### 2.1. Study Design

This was a prospective, multicentre, observational study recruiting individuals in close contact with index patients who received a new diagnosis of acid-fast smear (AFS)-positive pulmonary TB between September 2016 and August 2018. The study was conducted in two medical centres—the National Taiwan University Hospital in northern Taiwan and Kaohsiung Medical University Hospital in southern Taiwan—and their four branch hospitals. The study was approved by the institutional ethics committees of both medical centres (NTUH REC 201609044RINB and KMUHIRB-G[II]-20170033). 

### 2.2. Study Population

Individuals were eligible for enrollment if they were (1) aged ≥12 years; (2) in close contact with patients diagnosed with AFS-positive pulmonary TB; and (3) diagnosed with LTBI using either a tuberculin skin test (TST) or QuantiFERON-TB Gold in-tube assay (QFT; Cellestis/Qiagen, Carnegie, Australia). Close contact was defined as unprotected exposure of ≥8 h in a single day or a cumulative duration of ≥40 h. A positive TST was defined as an induration of ≥10 mm read at 48–72 h according to current guidelines in Taiwan. QFT was performed according to the manufacturer’s instructions. All close contacts enrolled in this study were tested with either TST or QFT. This study excluded participants who were suspected to have active TB because of their clinical symptoms or image examinations; to be concurrently using drugs with severe drug–drug interactions; to be allergic to INH, rifampin, or RPT (https://www.micromedexsolutions.com/); or who had a life expectancy <3 years.

### 2.3. Protocol

After written informed consent was obtained, participants were enrolled into the pharmacokinetics cohort (PK cohort) if they could comply with the blood sampling schedule required in the PK study. For this cohort, a plasma sample was collected to determine the concentrations of RPT, INH, and their metabolites (25-desacetyl-rifapentine [DeAcRPT] and acetyl-isoniazid [AcINH]) at either 23–25 h (C24, preferred) or 5–7 h (C6, *T*_max_ of RPT) or both after administration of the study drugs at weeks 4 and 8 (refer to Appendix A for laboratory methods, including Appendix A) [15,16], or while SDR developed. For participants not enrolled in the PK cohort, genotyping for single-nucleotide polymorphisms (SNPs) of drug-metabolising enzymes, including *N-acetyltransferase 2* (*NAT2*), *cytochrome P450 2E1* (*CYP2E1*), and *arylacetamide deacetylase* (*AADAC*), was performed at baseline (SNP cohort) (refer to Appendix A for laboratory method, including Appendix A).

Under supervision, each participant received weekly RPT (900 mg for participants with body weight >50.0 kg; 750 mg for 32.1–50.0 kg; 600 mg for 25.1–32.0 kg; and 450 mg for 14.1–25.0 kg) plus INH (15 mg/kg, rounded up to nearest 150 mg; maximum 900 mg) for a total of 12 doses, constituting the 3HP regimen. The drugs were administered by government-paid supporters under the National TB Program of Taiwan. The research assistants contacted all participants every week in person or by telephone to inquire about any AEs after the treatment. The participants were followed up until treatment completion or termination.

### 2.4. Outcome

The primary endpoint was development of SDRs during 3HP treatment, defined as AEs that met either of the following: (1) hypotension (systolic blood pressure < 90 mmHg), urticaria, angioedema, acute bronchospasm, or conjunctivitis; and (2) >4 of the following symptoms occurring concurrently (>1 of which had to be grade 2 or higher): weakness, fatigue, nausea, vomiting, headache, fever, aches, sweats, dizziness, shortness of breath, flushing, or chills [14]. The probability of AEs to the study drugs was determined using the Naranjo algorithm [17]. A Naranjo score of 5–8 indicates probable AEs, whereas a score of >9 indicates definite AEs. AEs were graded using the Division of AIDS Table for Grading the Severity of Adult and Pediatric Adverse Events [18].

### 2.5. Statistical Analysis

Collected data included age, sex, height, weight, smoking status, comorbidities, concurrent medication, laboratory test and imaging results, study drugs, AEs, and medical records. The dataset was independently sampled. For continuous variables with normal distribution and homogeneity of variance, we used independent-sample *t* test or one-way ANOVA for comparison of intergroup differences. Otherwise we used non-parametric methods including Mann–Whitney *U* test or Kruskal–Wallis test for comparison. For categorical variables, we used chi-square for intergroup comparison and in case that more than 20% of the expected cell counts for the table are less than five, we used Fisher’s exact test. Plasma drug concentration after 1 and 2 months of 3HP treatment was compared using the Wilcoxon rank-sum test. To accommodate correlation of values within subjects, generalised estimating equation (GEE) models were fitted to explore the risk factors of SDRs (binary outcome) during 3HP treatment, adjusting for age, sex, body mass index (BMI), smoking status, INH and RPT dosage (which was exactly the same), taking 3HP before or after meal, comorbidity and concomitant medications. We also performed time-dependent Cox proportional hazard model to explore SDRs risk factors with adjustment of abovementioned variables in the GEE model. For handling the issue of low extrapolated drug concentration level below limit of quantification (BLQ) (in Appendix A), we performed two sensitivity analyses, including: 1. Assuming drug concentrations BLQ to be zero and; 2. Excluding drug concentrations BLQ. Statistical significance was set at two-sided *p*  < 0.05. All statistical analyses were performed using SAS version 9.4 (SAS Institute Inc., Cary, NC, USA).

## 3. Results

### 3.1. Selection of Study Participants

During the study period, 177 participants were recruited to the SNP cohort and 129 to the PK cohort (Figure 1). The clinical characteristics of the two cohorts are summarised in Table 1.

### 3.2. Clinical Characteristics of the SNP Cohort

Among the 177 participants in the SNP cohort, 14 (8%) developed an SDR; seven were flu-like syndrome, three were shock, three were urticaria, and the remaining SDR was conjunctivitis. The majority of the participants were young (mean age 37.1 ± 17.8 years) and had good nutritional status (mean BMI 23.2 ± 3.52). Few (*n* = 11, 6%) had an underlying comorbidity, and the laboratory data were generally within normal limits. Compared with the participants who developed an SDR, those who did not were younger (*p* = 0.038) and had superior renal function (*p* = 0.009). 

### 3.3. Association of SNPs with SDRs in the SNP Cohort

In the SNP cohort, two SNPs were significantly associated with SDR development (Table 2). In a recessive model, *NAT2* rs1041983 was associated with SDR development in both univariate analysis (TT vs. CC+CT, odds ratio [OR] [95% CI]: 8.47 [2.55–28.1], *p* = 0.0005) and multivariate analysis (TT vs. CC+CT, OR [95% CI]: 7.00 [2.03–24.1], *p* = 0.002). Additionally, *CYP2E1* rs2070673 was associated with SDR development in both univariate analysis (AA vs. TT+TA, OR [95% CI]: 3.51 [1.05–11.7], *p* = 0.041) and multivariate analysis (AA vs. TT+TA, OR [95% CI]: 3.50 [1.02–12.0], *p* = 0.047). The results of SNPs with nonsignificant associations with SDRs are summarised in Appendix A. 

### 3.4. Clinical Characteristics of the PK Cohort

Among the 129 participants in the PK cohort, 13 (10.1%) developed an SDR (shock: two; flu-like syndrome: eight; both: three) (Table 1). The treatment completion rate was 83% (107/129). A total of 52 participants with 84 C6 blood samples were available, of which four samples were collected while they were experiencing an SDR. In C24 blood sampling, 144 samples from 83 participants were available; 11 samples were collected during an SDR. Among the 13 cases experiencing SDRs, ten (77%) occurred in the 3rd dose, 11 (85%) had the onset of SDRs less than 6 h after 3HP administration, and ten (77%) completely recovered within 2 days after onset (Table 3). 

Among the C6 blood samples, 50 samples were collected in the first month, whereas 34 were collected in the second month. Three SDRs developed among the participants who supplied 50 first-month C6 blood samples, whereas one SDR developed among the participants who supplied the 34 s-month blood samples (6% vs. 3%, *p* = 0.644). Among the C24 blood samples, 80 samples were collected in the first month, whereas 64 were collected in the second month. Seven SDRs developed among the participants who supplied the 80 first-month C24 blood samples, whereas four SDRs developed among the participants who supplied the 64 s-month blood samples (9% vs. 6%, *p* = 0.755). The correlations between plasma concentrations of drugs and metabolites are illustrated in Figure 2.

### 3.5. Pharmacodynamical Results of the PK Cohort

Regarding C6 drug concentrations, no differences were discovered between the participants with and without an SDR in INH, AcINH, RPT, or DeAcRPT (INH: 3.2 [2.4–3.9] vs. 2.0 [1.2–3.5] µg/mL, *p* = 0.663; RPT: 22.2 [18.8–27.8] vs. 20.7 [16.7–31.9] µg/mL, *p* = 0.832; Figure 3). Regarding C24 drug concentrations, INH level was significantly higher in the participants with an SDR compared with those without (0.25 [0.06–0.53] vs. 0.06 [0.05–0.15] µg/mL, *p* = 0.024). RPT level was not significantly different between the SDR and SDR-free participants (*p* = 0.184) in the C24 samples.

In C24 blood sampling, plasma INH level was higher among age >50 compared with age 30–50 and age <30. C24 RPT level, C6 INH level, and C6 RPT level, however, were not different between different age groups (Appendix A).

The C6 drug concentrations of AcINH and C24 drug concentrations of INH, AcINH, RPT, and DeAcRPT were significantly different between participants with differing renal function, being the highest in those with eGFR < 60 (Appendix A). 

The plasma levels of INH, RPT and their metabolites were not significantly altered by taking 3HP before or after meal, except that the C6 RPT level was significantly higher in participants taking 3HP after meal (*p* = 0.038) (Appendix A).

Comparing PK results for the first-month and second-month C6 blood samples (*n* = 32), the concentrations of INH, AcINH, RPT, and DeAcRPT were not significantly different (Appendix A). 

### 3.6. Sex-Based Discrepancy in PK Results

The drug dosage of INH and RPT was higher in the women than in the men (*p* < 0.0001; Figure 4).

In the C24 blood samples from 83 participants (44 male and 39 female), RPT level was higher in the women than in the men (*p* < 0.0001), whereas INH level was not (*p* = 0.536) (Figure 4). In C6 blood samples from 52 participants (21 male and 31 female), RPT level was again higher in the women than in the men (*p* = 0.018), whereas INH level was not (*p* = 0.549).

### 3.7. GEE Model in the PK Cohort

In the GEE model constructed for analysis of the C24 data, plasma INH level was associated with a higher risk of SDR development (OR [95% CI]: 1.61 [1.15–2.25], *p* = 0.006). By contrast, plasma RPT level was not associated with a higher risk of SDR development (OR [95% CI]: 1.01 [1.00–1.02], *p* = 0.218). In analysis assuming drug concentrations BLQ to be zero (OR [95% CI]: 1.52 [1.13–2.05], *p* = 0.006) and excluding drug concentration BLQ (OR [95% CI]: 1.94 [1.32–2.87], *p* = 0.001), the association of INH and SDRs remains. 

In analysis of the C6 data, no factors were significantly associated with SDR development in the GEE model (INH, OR [95% CI]: 1.00 [0.98–1.02], *p* = 0.990; RPT, OR [95% CI]: 1.00 [0.99–1.01], *p* = 0.996). 

### 3.8. Time-Dependent Cox Proportional Hazard Model in the PK Cohort

In the time-dependent Cox regression model for C24 data analysis in the PK cohort, plasma INH level remained associated with a higher risk of SDR development (HR [95% CI]: 39.2 [1.19–1291.4], *p* = 0.040). Plasma RPT level was not associated with a higher risk of SDR development (OR [95% CI]: 1.08 [0.86–1.36], *p* = 0.513).

In analysis of the C6 data, no factors were significantly associated with SDR development in the time-dependent Cox regression model. 

### 3.9. Validation for Predicting SDR Using SNPs

The *NAT2*/*CYP2E1* SNPs were validated in the PK cohort (Table 4). In a recessive model, *NAT2* rs1041983 was associated with SDR development in both univariate analysis (TT vs. CC+CT, OR [95% CI]: 4.23 [1.30–13.8], *p* = 0.017) and multivariate analysis (TT vs. CC+CT, OR [95% CI]: 4.43 [1.30–15.1], *p* = 0.017). *CYP2E1* rs2070673 was not associated with SDR development in either univariate analysis (AA vs. TT+TA, OR [95% CI]: 1.84 [0.46–7.41], *p* = 0.392) or multivariate analysis (AA vs. TT+TA, OR [95% CI]: 1.90 [0.46–7.80], *p* = 0.375). 

## 4. Discussion

This is the first prospective study investigating the effects of plasma INH and RPT levels and the SNPs of their metabolising enzymes on the risk of SDRs during 3HP therapy. We discovered that *NAT2* and probably *CYP2E1*, but not *AADAC*, gene SNPs were associated with the development of SDRs among individuals receiving the 3HP regimen. Interestingly, the plasma INH level, but not RPT level, was associated with SDR development in the PK study. 

The 3HP regimen is among the four regimens for LTBI that is recommended by the WHO and is also probably the most promising regimen because of its convenience, with only 12 doses required [19]. With its effectiveness well established, the major remaining concern regarding this latest LTBI regimen may be its AEs. Studies have estimated that 4.9% to 9.1% of those in close contact with patients with TB and who received 3HP failed to complete the regimen because of the side effects [10,12]. SDRs while on the 3HP regimen have generally been linked with RPT, which has a well-known side effect: flu-like syndrome (Appendix A) [14,20,21,22]. Additionally, RPT is a newer agent, making it a possible contributor to SDRs given that a higher SDR rate was observed compared with other INH-containing regimens. Some scholars, however, have argued against this point. First, one study using the 3HP regimen demonstrated that rechallenge with RPT did not necessarily lead to SDRs [14]. In the same study, rifapentine was better tolerated than isoniazid upon rechallenge [14]. In another study involving 1200 mg of RPT once weekly as continuation therapy for active TB, no SDRs were linked with RPT [23]. Finally, in a study of 162 pulmonary TB patients receiving RPT with a dosage of more than 15 mg/kg daily, no patients developed SDRs [24]. In our systematic review, we discovered that among reports describing an association between RPT and flu-like symptoms, RPT was commonly coadministered with INH (Appendix A). Furthermore, in cases describing an association between INH and flu-like symptoms, the associations were all proven with rechallenge (Appendix A) [25,26].

Although a less well-known effect than that of RPT, INH can also lead to flu-like syndrome (Appendix A) [25,26]. Of the patients with active TB who were receiving INH, usually with a dose of 300 mg/day, 1–9.8% developed flu-like syndrome. In the 3HP regimen, INH dosage is 900 mg/week, and no data exist regarding the proportion of cases developing SDRs under a single INH dosage higher than this.

The association between the high INH dose and SDR during 3HP therapy may be explained through some hypotheses. First, since INH could bind to key enzymes in cytokine pathways, such as peroxidase [27], a high plasma level of INH may activate pathways which are not activated under normal dose of INH due to low binding affinity. Second, the high plasma INH level may interfere with or interact with rifapentine metabolite, leading to SDR. Interestingly, a study investigating drug–drug interactions between dolutegravir and once weekly INH/RPT also revealed a higher INH area under concentration curve among those who develop toxicities and a higher INH level among two cases experiencing severe flu-like syndrome [28].

NAT2 protein metabolises INH into AcINH, which is later hydrolysed into acetylhydrazine. Acetylhydrazine is then oxidised by CYP2E1 to form a hepatotoxic substance, causing damage to hepatocytes [29]. Additionally, AADAC is the enzyme responsible for the deacetylation of rifamycins, and it affects the metabolism of rifamycins [30]. In our study, several points support the association between INH and SDR development. First, the *NAT2* SNP, which is known to affect INH metabolism, rather than the *AADAC* SNPs, which are involved in RPT metabolism, was associated with SDR development. Second, INH drug level, rather than RPT drug level, was discovered to be associated with SDR development. Third, the short duration and rapid resolution of symptoms in some cases may indicate that a rapidly metabolised drug was the causative agent. 

In our study, there was a sex-based discrepancy in the RPT plasma level but not in INH level. Sex differences in pharmacokinetics and pharmacodynamics for many drugs have been documented before and were commonly attributed to the endogenous hormone influence on cytochrome P450 activity [31]. Our study observed a higher RPT level in female and this phenomenon was not observed for INH. This may be explained that INH level was more significantly modulated and determined by acetylation rate [29]. In previous study, the proportion of fast, slow and intermediate acetylators was not different between male and female [32]. 

For practical clinical application of our study findings, one may consider testing *NAT2* genotype before LTBI treatment. If our described SNPs are identified, LTBI cases may be suggested to receive preventive regimens other than 3HP, such as four-month rifampin, to avoid SDR [33]. Also, for developing better drug combination therapy for LTBI, tailoring isoniazid dosage by avoiding single large isoniazid dose may be a reasonable approach. In the recently published Brief TB trial comparing one-month isoniazid and rifapentine with nine-month INH for preventive therapy, the isoniazid dosage was reduced to 300 mg daily and SDRs seemed to be less observed [34].

Our study also had limitations. First, the sample size was relatively small. Therefore, we may not have been able to identify all the SNPs associated with SDRs. Second, this study was conducted in the Taiwanese population. In our previous study, the proportion of NAT2 slow-acetylators was 22.8% (*n* = 82) among 360 TB patients [35]. Whether our findings can be extrapolated to other ethnicities remains unknown.

## 5. Conclusions

Results of this study suggest that INH plays a more critical role than is generally perceived in 3HP-related SDRs. By designing a more optimal LTBI regimen, this study highlights the importance of INH in causing SDRs. Also, *NAT2* SNP could also be used for risk stratification among TB contacts receiving 3HP regimen.

## Figures and Tables

**Figure 1 jcm-08-00812-f001:**
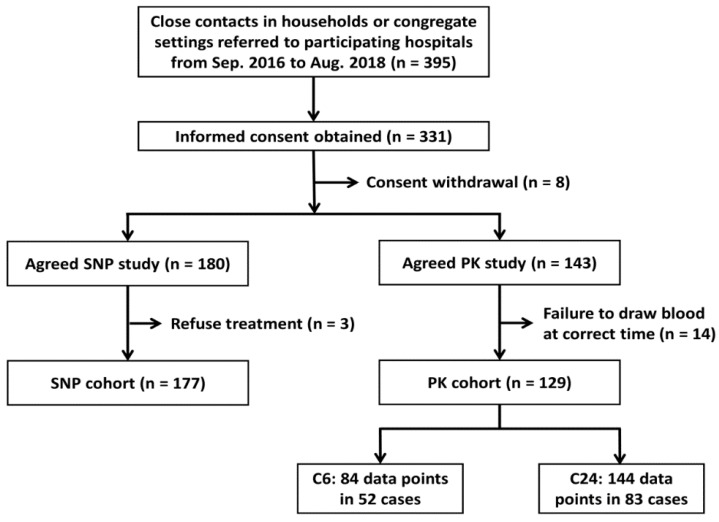
Case enrollment (PK: pharmacokinetic; SNP: single-nucleotide polymorphism).

**Figure 2 jcm-08-00812-f002:**
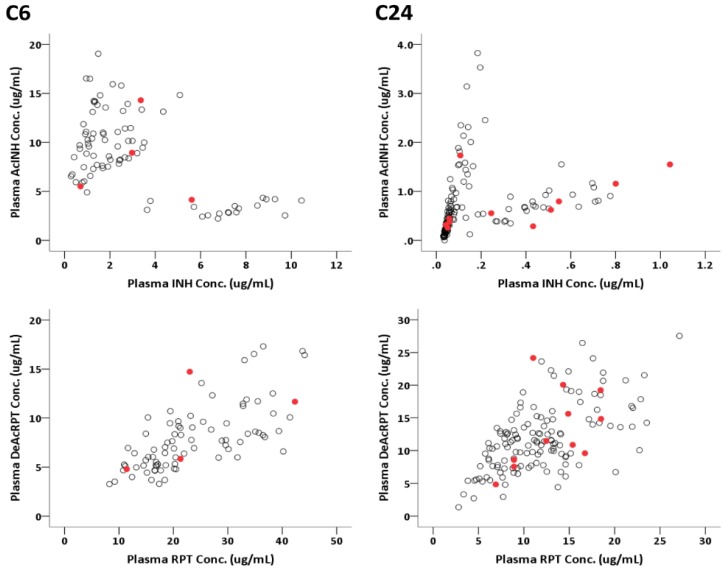
XY plot showing the correlations between plasma concentration of isoniazid (INH) and acetyl-isoniazid (AcINH) (upper panels), and plasma concentration of rifapentine (RPT) and desacetyl-rifapentine (DeAcPRT) (lower panels) at 6 (C6, left column) and 24 (C24, right column) hours after drug administration, with cases who experienced a systemic drug reaction (SDR) marked in red.

**Figure 3 jcm-08-00812-f003:**
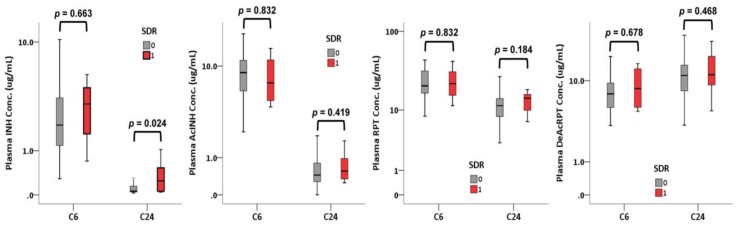
Boxplot showing the concentration of isoniazid (INH), acetyl-isoniazid (AcINH), rifapentine (RPT), and desacetyl-rifapentine (DeAcRPT) at 6 (C6) and 24 (C24) hours after dosing, stratified by the development of systemic drug reactions (SDRs) (*p* value calculated using the Mann–Whitney *U* test).

**Figure 4 jcm-08-00812-f004:**
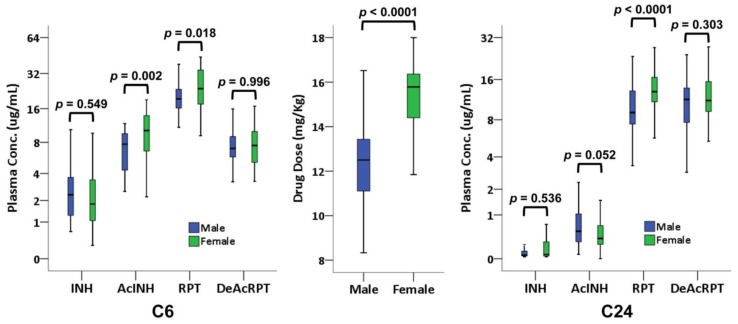
Boxplot showing drug dosage per kg body weight (middle panel) and plasma concentration of INH, acetyl-isoniazid (AcINH), rifapentine (RPT), and desacetyl-rifapentine (DeAcRPT) in male and female participants at 6 (C6, left panel) and 24 (C24, right panel) hours after dosing (*p* value calculated using the Mann–Whitney *U* test).

**Table 1 jcm-08-00812-t001:** Clinical characteristics and laboratory data.

	SNP Group (*n* = 177)	SDR (*n* = 14)	No SDR (*n* = 163)	*p* Value	PK Group (*n* = 129)	SDR (*n* = 13)	No SDR (*n* = 116)	*p* Value
Age (year)	37.1 ± 17.8	46.6 ± 14.5	36.3 ± 17.9	0.038	48.8 ± 17.2	51.6 ± 12.7	48.5 ± 17.6	0.533
≤35	94 (53%)	2 (14%)	92 (56%)	0.002	30 (23%)	2 (15%)	28 (24%)	0.868
35–55	44 (25%)	8 (57%)	36 (22%)		46 (36%)	5 (38%)	41 (35%)	
>55	39 (22%)	4 (29%)	35 (21%)		53 (41%)	6 (46%)	47 (41%)	
Female sex	83 (47%)	6 (43%)	77 (47%)	0.753	67 (52%)	6 (46%)	61 (53%)	0.660
Household contact	38 (21%)	2 (13%)	36 (22%)	0.737	53 (41%)	5 (38%)	48 (41%)	>0.999
Height (cm)	165.8 ± 8.3	165.1 ± 8.8	165.9 ± 8.2	0.729	164.3 ± 9.0	163.7 ± 7.5	164.4 ± 9.1	0.788
Weight (kg)	64.0 ± 11.9	65.7 ± 12.0	63.8 ± 10.9	0.568	65.5 ± 12.1	63.1 ± 9.4	65.8 ± 12.4	0.444
Body-mass index (kg/m^2^)	23.2 ± 3.52	24.1 ± 3.22	23.1 ± 3.55	0.334	24.2 ± 3.36	23.45 ± 2.23	24.26 ± 3.46	0.413
Current smoker	20 (11%)	4 (29%)	16 (80%)	0.057	28 (22%)	5 (38%)	23 (20%)	0.154
eGFR (mL/min/1.73 m^2^)				0.005				0.670
<60	14 (8%)	0	14 (9%)		6 (5%)	1 (8%)	5 (4%)	
60–90	54 (31%)	10 (71%)	44 (27%)		47 (36%)	4 (31%)	43 (37%)	
≥90	109 (62%)	4 (29%)	105 (66%)		77 (59%)	8 (62%)	68 (59%)	
Comorbidity								
HBV infection	3 (2%)	0	3 (2%)	>0.999	7 (5%)	1 (8%)	6 (5%)	0.534
HCV infection	2 (1%)	0	2 (1%)	>0.999	3 (2%)	0	3 (3%)	>0.999
Diabetes mellitus	3 (2%)	0	3 (2%)	>0.999	11 (9%)	2 (15%)	9 (8%)	0.306
Malignancy	1 (1%)	1 (7%)	0	0.079	6 (5%)	2 (15%)	4 (3%)	0.112
Autoimmune	1 (1%)	0	1 (1%)	>0.999	1 (1%)	1 (8%)	0	0.100
Asthma	0	0	0		1 (1%)	0	1 (1%)	>0.999
Hypertension	5 (3%)	2 (14%)	3 (2%)	0.051	25 (19%)	5 (38%)	20 (17%)	0.130
Anti-hypertensive medication	5 (3%)	2 (14%)	3 (2%)	0.051	19 (15%)	4 (31%)	15 (13%)	0.101
Isoniazid dose (mg/kg)	14.2 ± 2.1	13.8 ± 2.0	14.3 ± 2.1	0.483	14.0 ± 2.2	14.3 ± 1.9	13.9 ± 2.2	0.512
Rifapentine dose (mg/kg)	14.2 ± 2.1	13.8 ± 2.0	14.3 ± 2.1	0.454	14.0 ± 2.2	14.3 ± 1.9	13.9 ± 2.2	0.512
Hemoglobin (g/dL)	14.0 ± 1.6	14.2 ± 1.5	14.0 ± 1.6	0.643	14.0 ± 1.5	13.8 ± 1.6	14.1 ± 1.5	0.560
Leukocyte (K/µL)	6.44 ± 1.77	6.78 ± 1.42	6.41 ± 1.80	0.448	6.81 ± 1.85	6.98 ± 1.44	6.78 ± 1.90	0.732
Platelet (K/µL)	258 ± 56	253 ± 57	259 ± 56	0.705	270 ± 58	280 ± 45	269 ± 59	0.511
AST (U/L)	23.4 ± 17.0	28.0 ± 19.6	23.0 ± 16.8	0.291	23.3 ± 10.0	25.5 ± 5.6	23.0 ± 10.4	0.201
ALT (U/L)	23.0 ± 28.0	27.6 ± 30.8	22.6 ± 27.9	0.526	23.7 ± 18.9	27.2 ± 11.1	23.3 ± 19.6	0.290
Total bilirubin (mg/dL)	0.65 ± 0.28	0.63 ± 0.38	0.66 ± 0.27	0.823	0.63 ± 0.22	0.70 ± 0.25	0.62 ± 0.22	0.215
Creatinine (mg/dL)	0.82 ± 0.20	0.83 ± 0.16	0.82 ± 0.20	0.754	0.84 ± 0.29	0.83 ± 0.18	0.84 ± 0.30	0.876
Treatment completion	159 (90%)	4 (29%)	155 (95%)	<0.0001	107 (83%)	4 (31%)	103 (89%)	<0.0001

ALT: alanine transaminase; AST: aspartate transaminase; eGFR: estimated glomerular filtration rate; PK: pharmacokinetic; SDR: systemic drug reaction; SNP: single nucleotide polymorphism. Data are number (percentage) or mean ± standard deviation.

**Table 2 jcm-08-00812-t002:** Association of *NAT2*/*CYP2E1* single-nucleotide polymorphisms (SNPs) with systemic drug reactions.

		Unadjusted OR (95% CI)	*p* Value	Adjusted OR (95% CI) *	*p* Value
**Additive model**					
*NAT2* rs1041983	CC	Ref		Ref	
	CT	0.85 (0.14–5.29)	0.101	0.87 (0.14–5.46)	0.132
	TT	7.67 (1.51–39.0)	0.0006	5.82 (1.08–35.1)	0.003
*CYP2E1* rs2070673	TT	Ref		Ref	
	TA	0.84 (0.20–3.52)	0.815	0.89 (0.21–3.80)	0.871
	AA	3.21 (0.79–15.0)	0.103	3.28 (0.78–13.9)	0.106
**Dominant model**					
*NAT2* rs1041983	CC	Ref		Ref	
	CT+TT	2.41 (0.51–11.3)	0.265	2.01 (0.41–9.96)	0.394
*CYP2E1* rs2070673	TT	Ref		Ref	
	TA+AA	1.43 (0.42–4.84)	0.568	1.49 (0.43–5.20)	0.532
**Recessive model**					
*NAT2* rs1041983	CC+CT	Ref		Ref	
	TT	8.47 (2.55–28.1)	0.0005	7.00 (2.03–24.1)	0.002
*CYP2E1* rs2070673	TT+TA	Ref		Ref	
	AA	3.51 (1.05–11.7)	0.041	3.50 (1.02–12.0)	0.047

* Adjusted for age, sex and estimated glomerular filtration rate.

**Table 3 jcm-08-00812-t003:** Detailed information of the participants experiencing systemic drug reaction during 3HP therapy in the pharmacokinetic (PK) cohort.

Age/Sex	BW (kg)/BH (cm)	Adverse Reactions	Severity (Grade)	Comorbidity & Medication	Risk Allele in *NAT2*/*CYP2E1* *	INH/RPT Conc. ^#^ [Sampling Week]	Onset (Week)	Time of Onset/Duration (h)	Outcome of 3HP
66.3/F	50.0/149	fever, chills, malaise, myalgia, headache	2	Lung adenocarcinoma under gefitinib	1/2	**C24: 0.11**/**18.5** [3]	3	7/29	Stop
64.4/M	62.5/170	fever, myalgia, chills, weakness, sweating	2	HTN under amlodipine & olmesartan	2/1	**C6: 5.61**/**21.3** [4]	3	6/18	Stop
59.8/F	59.5/154	fever, chills, dyspnea, angioedema, malaise	3	Breast cancer, cured	2/2	**C24: 0.51**/**16.7** [3]	3	4/>100	Stop
56.7/M	73.0/175	shock (BP 90/60 mmHg), fever, flush, myalgia, dyspnea, rash	3	HTN under lercanidipine	2/2	**C24: 1.04**/8.9 [3]	3	5/47	Stop
53.4/F	63.0/160	fever, chills, dizziness, myalgia, dizziness	2	Nil	1/1	**C6: 2.98**/11.4 [4]	3	9/15	Stop
51.4/M	74.0/167	shock (BP 85/67 mmHg), dizziness, vomiting	2	DM, HTN	2/1	**C24: 0.80**/8.9 [3]	3	3/47	Stop
50.5/M	72.0/168	shock (BP 88/63 mmHg), fever, nausea, vomiting, dizziness, sweating	2	HTN under bisoprolol & olmesartan	2/0	**C24: 0.25**/6.9 [7]	7	1/8	Stop
33.9/F	47.0/158	shock (BP 82/57 mmHg), fever, headache, nausea, vomiting, malaise	3	Nil	2/0	**C24: 0.43**/**15.4** [3]	3	1/88	Stop
20.6/F	53.0/162	shock, fever, chills, headache, myalgia, nausea	3	Nil	1/1	**C6: 3.36**/**42.3** [4] **C24: 0.06**/**18.4** [4]	3	2/30	Stop
60.9/M	62.0/161	fever, myalgia, nausea, vomiting dizziness	2	DM, HTN amlodipine & valsartan	0/1	**C24: 0.06**/11.0 [3]	3	6/28	Complete
55.9/F	60.0/163	fever, chills, myalgia, malaise, headache	3	AS under celecoxib	0/0	C24: 0.04/**14.3** [4]	3	5/76	Complete
53.7/M	76.5/174	fever, chills, dizziness, malaise, nausea	2	HBV carrier not Tx	2/1	**C24: 0.55**/**14.9** [6]	6	3/16	Complete
43.8/M	67.5/167	fever, myalgia, dizziness, tachypnea, malaise	2	Nil	1/0	C6: 0.70/**23.0** [4]	4	1/27	Complete

AS, ankylosing spondylitis; BP, blood pressure; CYP2E1, cytochrome P450 2E1; DM, diabetes mellitus; HTN, hypertension; INH, isoniazid; NAT2, *N-acetyltransferase 2*, RPT, rifapentine; Tx, treatment; * Number of T allele at *NAT2* rs1041983/number of A allele at *CYP2E1* rs2070673; ^#^ Drug levels that were higher than the median of all tested data at the same timing (C6 of INH: 2.07 µg/mL; C6 of RPT: 20.9 µg/mL; C24 of INH: 0.06 µg/mL; C24 of RPT: 11.4 µg/mL) were shown in bold.

**Table 4 jcm-08-00812-t004:** Validation of *N-acetyltransferase 2* (*NAT2*)/*Cytochrome P450 2E1* (*CYP2E1*) single nucleotide polymorphisms (SNPs) with systemic drug reactions in pharmacokinetic (PK) cohort.

		Unadjusted OR (95% CI)	*p* Value	Adjusted OR (95% CI) *	*p* Value
**Additive model**					
*NAT2* rs1041983	CC	Ref		Ref	
	CT	1.09 (0.19–6.28)	0.925	1.06 (0.18–6.34)	0.948
	TT	4.52 (0.86–23.8)	**0.075**	4.61 (0.82–25.8)	0.082
*CYP2E1* rs2070673	TT	Ref		Ref	
	TA	1.84 (0.49–6.94)	0.807	2.10 (0.54–8.20)	0.285
	AA	2.53 (0.51–12.5)	0.383	2.80 (0.55–14.3)	0.216
**Dominant model**					
*NAT2* rs1041983	CC	Ref		Ref	
	CT+TT	2.13 (0.45–10.2)	0.343	2.04 (0.41–10.1)	0.384
*CYP2E1* rs2070673	TT	Ref		Ref	
	TA+AA	2.03 (0.59–6.96)	0.262	2.30 (0.65–8.15)	0.199
**Recessive model**					
*NAT2* rs1041983	CC+CT	Ref		Ref	
	TT	4.23 (1.30–13.8)	**0.017**	4.43 (1.30–15.1)	**0.017**
*CYP2E1* rs2070673	TT+TA	Ref		Ref	
	AA	1.84 (0.46–7.41)	0.392	1.90 (0.46–7.80)	0.375

* Adjusted for age, sex and estimated glomerular filtration rate.

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
