# Peer review of "Isoniazid Concentration and *NAT2* Genotype Predict Risk of Systemic Drug Reactions during 3HP for LTBI"

_jcm, 2019, doi:10.3390/jcm8060812_

Round 1
Reviewer 1 Report
Authors address an important aspect of LTB drugs. I have 3 main concerns regarding the study that might improve this important work.
1. It is not clear from the manuscript if the individuals enrolled as ‘close contact’ were also tested for TST or QFT – if no, then this is a serious confounding issue and possibly the reason why authors observe such large confidence intervals in the odds ratio eg. (TT vs CC+CT 7.00[2.03 – 24.1]). It will be great if authors can clarify this as the study largely depends on the number of individuals enrolled.
2. It might also be unethical to enroll individuals based on ‘close contact’ and conduct the study. The definition of ‘close contact’ requires an in-depth explanation along with the evidence of TST or QFT tests.
3. On Statistical method – since it is a prospective observational study, therefore it might be worth trying Cox-proportional hazard modeling to infer the hazard ratios instead of odds ratios. Cox-proportional hazard modeling will allow adjusting for comorbidities including demographics. I personally feel that the hazard ratios are more meaningful than the odds ratio in this current scope of the study.
Author Response
Authors address an important aspect of LTB drugs. I have 3 main concerns regarding the study that might improve this important work.
1. It is not clear from the manuscript if the individuals enrolled as ‘close contact’ were also tested for TST or QFT – if no, then this is a serious confounding issue and possibly the reason why authors observe such large confidence intervals in the odds ratio eg. (TT vs CC+CT 7.00[2.03 – 24.1]). It will be great if authors can clarify this as the study largely depends on the number of individuals enrolled.
Ans: Thanks for the excellent comments. All close contacts were tested for TST or QFT. We have added a description in this study.
“A positive TST was defined as an induration of ≥ 10 mm read at 48–72 hours according to current guideline in Taiwan. QFT was performed according to the manufacturer’s instructions. All close contacts enrolled in this study were tested with either TST or QFT.”
2. It might also be unethical to enroll individuals based on ‘close contact’ and conduct the study. The definition of ‘close contact’ requires an in-depth explanation along with the evidence of TST or QFT tests.
Ans: Thanks for the excellent suggestions. We have added the definition of close contacts.
“Close contact was defined as unprotected exposure of ≥8 hours in a single day or a cumulative duration of ≥40 hours.”
3. On Statistical method – since it is a prospective observational study, therefore it might be worth trying Cox-proportional hazard modeling to infer the hazard ratios instead of odds ratios. Cox-proportional hazard modeling will allow adjusting for comorbidities including demographics. I personally feel that the hazard ratios are more meaningful than the odds ratio in this current scope of the study.
Ans: Thanks for the excellent suggestions. We have also performed time-dependent Cox proportional hazard model accordingly. The results were also described. Due to the complicated and different timing of sampling (C6, C24 and also at weeks 4 and weeks 8), we still used GEE model as main analysis and added the results of time-dependent Cox proportional hazard model in the results section.
“We also performed time-dependent Cox proportional hazard model to explore SDRs risk factors with adjustment of abovementioned variables in the GEE model.”
“ 3.8. Time-dependent Cox proportional hazard model in the PK cohort
In time-dependent Cox regression model for C24 data analysis in the PK cohort, plasma INH level remained associated with a higher risk of SDR development (HR [95% CI]: 39.2 [1.19–1291.4], p = 0.040). Plasma RPT level was not associated with a higher risk of SDR development (OR [95% CI]: 1.08 [0.86–1.36], p = 0.513).In analysis of the C6 data, no factors were significantly associated with SDR development in the time-dependent Cox regression model.”
Reviewer 2 Report
Summary:
The authors provide evidence for associations between systemic drug reactions and the concentration of a commonly used anti-TB drug isoniazid, as well as single nucleotide polymorphisms. This provides an important framework from which improved drug selection protocols could be developed for personalized medicine of patients with LTBI.
Limitations and strengths:
The authors should be congratulated on their discovery of factors that are linked with SDRs during 3HP therapy, which will provide the opportunity to choose treatment regimes with less adverse effects for patients with specific SNPs. The study is thorough and the manuscript reads very well, especially the logical presentation of the findings, and therefore there are only a few aspects which I believe need improving before publication.
The small study size and homogenous cohort ethnicity are limitations of the study, though these have both been acknowledged in the discussion. Other aspects of the broader implications of the results could be described more thoroughly in the discussion, as detailed below.
Specific items:
To improve the completeness and ability for readers to understand the results and rational for the experimental design, please add a brief statementin appropriate section of the manuscript, explaining/discussing/rectifying the following items:
· The righthand portion of Table 3 appears to be clipped, which should be rectified by the authors/MDPI formatting staff before publication.
· Methods section 2.5. For completeness, please add further details about the data sets so that the choice of statistical tests can be justified. E.g. describe the variances, whether the data sets contained independent/unbiased samples, and whether they were normally distributed.
· In the discussion, speculate why the effects described in line 211 to 216 occur, e.g. why would men and women have different RPT levels, and not INH levels. Has this been documented with other drugs? Please add citations if available.
· A short explanation of how these results have been, or are likely to be used in the design of improved drug treatment combinations, currently, or in the future, is missing from the discussion section. E.g. is it feasible to identify SNPs in large numbers of people before choosing a LTBI therapy, considering the prevalence of TB in lesser developed countries? If the described SNPs are identified, what might be an approach for choosing a better drug combination and why? Please contextualize the current results with other risk-stratification measures that are used for TB or other highly prevalent infections/diseases where medium term prophylactic/therapeutic drug regimes are used. Please add citations if available.
Optional items:
· The introduction contains all of the background information necessary for the ready, however, it is very concise, and could be expanded at the authors discretion.
Author Response
The authors provide evidence for associations between systemic drug reactions and the concentration of a commonly used anti-TB drug isoniazid, as well as single nucleotide polymorphisms. This provides an important framework from which improved drug selection protocols could be developed for personalized medicine of patients with LTBI.
Limitations and strengths:
The authors should be congratulated on their discovery of factors that are linked with SDRs during 3HP therapy, which will provide the opportunity to choose treatment regimens with less adverse effects for patients with specific SNPs. The study is thorough and the manuscript reads very well, especially the logical presentation of the findings, and therefore there are only a few aspects which I believe need improving before publication.
The small study size and homogenous cohort ethnicity are limitations of the study, though these have both been acknowledged in the discussion. Other aspects of the broader implications of the results could be described more thoroughly in the discussion, as detailed below.
Ans: Thanks for the instructive comments. We have revised accordingly.
Specific items:
To improve the completeness and ability for readers to understand the results and rational for the experimental design, please add a brief statement in appropriate section of the manuscript, explaining/discussing/rectifying the following items:
· The righthand portion of Table 3 appears to be clipped, which should be rectified by the authors/MDPI formatting staff before publication.
Ans: Thanks for the suggestions. We have carefully checked and ensured that Table 3 is not clipped in our uploaded docx and pdf files.
Methods section 2.5. For completeness, please add further details about the data sets so that the choice of statistical tests can be justified. E.g. describe the variances, whether the data sets contained independent/unbiased samples, and whether they were normally distributed.
Ans: Thanks for the comments. We have revised accordingly and added more detailed descriptions in the method section.
“The dataset was independently sampled. For continuous variables with normal distribution and homogeneity of variance, we used independent-sample t test or one way ANOVA for comparison of intergroup differences. Otherwise we used non-parametric methods including Mann–Whitney U test or Kruskal–Wallis test for comparison. For categorical variables, we used chi-square for intergroup comparison and in case that more than 20% of the expected cell counts for the table are less than 5, we used Fisher’s exact test.”
·3. In the discussion, speculate why the effects described in line 211 to 216 occur, e.g. why would men and women have different RPT levels, and not INH levels. Has this been documented with other drugs? Please add citations if available.
Ans: Thanks for the suggestions. We have added a paragraph discussing and speculating why sex-based discrepancy was observed in female but not in male.
“In our study, there was a sex-based discrepancy in the RPT plasma level but not in INH level. Sex differences in pharmacokinetics and pharmacodynamics for many drugs have been documented before and were commonly attributed to the endogenous hormone influence on cytochrome P450 activity [31]. Our study observed a higher RPT level in female and this phenomenon was not observed for INH. This may be explained that INH level was more significantly modulated and determined by acetylation rate [29]. In previous study, the proportion of fast, slow and intermediate acetylators were not different between male and female [32].”
4. A short explanation of how these results have been, or are likely to be used in the design of improved drug treatment combinations, currently, or in the future, is missing from the discussion section. E.g. is it feasible to identify SNPs in large numbers of people before choosing a LTBI therapy, considering the prevalence of TB in lesser developed countries? If the described SNPs are identified, what might be an approach for choosing a better drug combination and why? Please contextualize the current results with other risk-stratification measures that are used for TB or other highly prevalent infections/diseases where medium term prophylactic/therapeutic drug regimens are used. Please add citations if available.
Ans: Thanks for the suggestions. We have added a paragraph discussing potential clinical application of our findings and added citations accordingly.
“For practical clinical application of our study findings, one may consider testing NAT2 genotype before LTBI treatment. If our described SNPs are identified, LTBI cases may be suggested to receive preventive regimens other than 3HP, such as four-month rifampin, to avoid SDR [33]. Also, for developing better drug combination therapy for LTBI, tailoring isoniazid dosage by avoiding single large isoniazid dose may be a reasonable approach. In the recently published Brief TB trial comparing one-month isoniazid and rifapentine with nine-month INH for preventive therapy, the isoniazid dosage was reduced to 300mg daily and SDRs seemed to be less observed [34].”
Optional items:
5. The introduction contains all of the background information necessary for the ready, however, it is very concise, and could be expanded at the authors discretion.
Ans: Thanks for the suggestions. We have expanded the introduction section.
“The importance of targeting LTBI toward TB control has extended from countries with low TB prevalence to TB-endemic areas [6, 7]. LTBI treatment, therefore, is being advocated as a universal policy in TB control [6, 7].”
“Also, for the 4 LTBI regimens currently suggested by WHO, 3HP remained the one with the fewest total doses required [13]. Though 3HP is less hepatotoxic, 3.8% of those receiving 3HP experience systemic drug reactions (SDRs) [14], which usually, if not always, requires treatment interruption or termination.”
Round 2
Reviewer 1 Report
I thank authors for their revised version of the article and feel that article may be accepted for the publication.